# Effect of Deep versus Moderate Neuromuscular Blockade on Quantitatively Assessed Postoperative Atelectasis Using Computed Tomography in Thoracic Surgery; a Randomized Double-Blind Controlled Trial

**DOI:** 10.3390/jcm10153228

**Published:** 2021-07-22

**Authors:** Bong-Jae Lee, Han Na Lee, Jun-Young Chung, Daehyun Kim, Jung Im Kim, Hyungseok Seo

**Affiliations:** 1Department of Anesthesiology and Pain Medicine, Kyung Hee University Hospital at Gangdong, College of Medicine, Kyung Hee University, Seoul 05278, Korea; lbj8350@naver.com (B.-J.L.); madsleep@naver.com (J.-Y.C.); 2Department of Radiology, Kyung Hee University Hospital at Gangdong, College of Medicine, Kyung Hee University, Seoul 05278, Korea; lhnangel@gmail.com (H.N.L.); mine147@gmail.com (J.I.K.); 3Department of Thoracic Surgery, Kyung Hee University Hospital at Gangdong, College of Medicine, Kyung Hee University, Seoul 05278, Korea; kmctskdh@hanmail.net

**Keywords:** atelectasis, deep block, neuromuscular blockade, deep block, postoperative pulmonary complications, sugammadax

## Abstract

Background: postoperative atelectasis is a significant clinical problem during thoracic surgery with one-lung ventilation. Intraoperative deep neuromuscular blockade can improve surgical conditions, but an increased risk of residual paralysis may aggravate postoperative atelectasis. Every patient was verified to have full reversal before extubation. We compared the effect of deep versus moderate neuromuscular blockade on postoperative atelectasis quantitatively using chest computed tomography. Methods: patients undergoing thoracic surgery were randomly allocated to two groups: moderate neuromuscular blockade during surgery (group M) and deep neuromuscular blockade during surgery (group D). The primary outcome was the proportion and the volume of postoperative atelectasis measured by chest computed tomography on postoperative day 2. The mean values of the repeatedly measured intraoperative dynamic lung compliance during surgery were also compared. Result: the proportion of postoperative atelectasis did not differ between the groups (1.32 [0.47–3.20]% in group M and 1.41 [0.24–3.07]% in group D, *p* = 0.690). The actual atelectasis volume was 38.2 (12.8–61.4) mL in group M and 31.9 (7.84–75.0) mL in group D (*p* = 0.954). Some factors described in the lung protective ventilation were not taken into account and might explain the atelectasis in both groups. The mean lung compliance during one-lung ventilation was higher in group D (26.6% in group D vs. 24.1% in group M, *p* = 0.026). Conclusions: intraoperative deep neuromuscular blockade did not affect postoperative atelectasis when compared with moderate neuromuscular blockade if full reversal was verified.

## 1. Introduction

Postoperative atelectasis can be a major cause of morbidity after non-cardiac surgery [1]. In thoracic surgery, one-lung ventilation (OLV) can achieve appropriate surgical field exposure in the ipsilateral lung. However, during OLV, the risk of pulmonary atelectasis can increase due to the use of a higher inspiratory oxygen fraction (FiO_2_) insufficient re-expansion of the collapsed lung and of the dependent lung [2]. Several strategies such as avoiding residual neuromuscular effects, appropriate analgesia, lung recruitment and low oxygen fraction before extubation under CPAP, avoiding opioids and sedatives peri operative to reduce obstructive breathing and deep breathing exercises can be applied to reduce atelectasis after thoracic surgery [3,4].

Neuromuscular blockade (NMB) can improve intubation conditions and surgical exposure, but it has also been associated with the risk of postoperative pulmonary complications (PPCs) due to residual muscle paralysis. Compared with moderate NMB, deep NMB improves laparoscopic surgical condition and decrease involuntary patient movement. However, deep NMB may also increase the incidence of residual paralysis and postoperative pulmonary complications [5,6]. However, the increased risk of PPCs due to residual NMB effect remains a relevant issue and the usefulness of intraoperative deep or moderate NMB is still controversial [7,8]. Recently, sugammadex was shown to allow rapid reversal of rocuronium-induced deep NMB and may prevent postoperative residual paralysis and subsequent atelectasis after OLV [5,7].

In the present study, we investigated the effect of deep versus moderate NMB with always full reversal on quantitatively measured postoperative atelectasis using chest computed tomography (CT) in patients undergoing thoracic surgery.

## 2. Materials and Methods

### 2.1. Study Design

In this randomized, double-blind, single-center clinical trial, we compared the effect of deep versus moderate NMB in OLV on postoperative atelectasis quantitatively using chest CT. The present study was approved by the Institutional Review Board of Kyung Hee University Hospital at Gangdong (approval number: KHNMC 2018-03-015-002). The study was registered before patient enrollment in the clinical research registry (https://clinicaltrials.gov, accessed on 11 September 2020, NCT03503565). This study was conducted between August 2018 and May 2020. Written informed consent was obtained from all participants before inclusion in the study.

### 2.2. Patients

Altogether, 118 adults with American Society of Anesthesiologists physical status 1–3 who were scheduled to undergo thoracic surgery with OLV (such as wedge resection, segmentectomy, and lobectomy) were enrolled. The exclusion criteria were: (1) patients with body mass index <18.5 kg/m^2^ or >35.0 kg/m^2^, (2) patients contraindicated for thoracic epidural catheter insertion for postoperative pain control, (3) patients anticipated to have an OLV of <60 min, (4) patients with uncontrolled diabetes mellitus, (5) severe renal dysfunction such as requirement of hemodialysis for end-stage renal disease, (6) neuromuscular diseases such as myasthenia gravis, (7) major burns (≥third degree burns), (8) compromised cardiopulmonary function, and (9) patients with current pregnancy or chance to be pregnant.

### 2.3. Allocation, Randomization and Blindness

All enrolled patients were randomized with a predetermined, computer-generated random assignment table using the random 4-block and 6-block technique. Patients were divided into groups M or D on the day of surgery by an independent investigator who did not participate in the entire anesthesia procedure. In group M, moderate NMB was maintained during the surgery while monitoring with a train-of-four (TOF) count of 1 or 2. In group D, deep NMB was performed intraoperatively while monitoring with post-tetanic count (PTC) of 1 or 2. The anesthesiologist was aware of each patient’s group, but was not involved in the outcome assessment. During surgery, the patient’s hand used to monitor the NMB was covered with drapes to maintain blinding of the surgeon. The patient and the radiologist who assessed postoperative CT scans were also blinded to the intervention.

### 2.4. Anesthesia Protocols

On arrival at the operating room, standard patient monitoring including electrocardiogram, peripheral oxygen saturation (SpO_2_), bispectral index, and non-invasive blood pressure monitoring was performed. Anesthesia was induced and maintained with a target-controlled infusion of propofol (Schnider model) and remifentanil (Minto model). Rocuronium bromide (0.6 mg/kg) was administered to facilitate tracheal intubation. Patients were manually ventilated with high flow (more than 8 L/min) of air/oxygen mixture (FiO_2_ 0.8) at 5 cmH_2_O of adjusted pressure limitation, and intubated with a double-lumen tracheal tube (37 Fr for men and 35 Fr for women) under videolaryngoscopy. In case of showing an obstruction breathing pattern on end-tidal carbon dioxide (ETCO_2_) curve during manual ventilation, oro-pharyngeal airway was used or head-position was changed. Tracheal intubation was confirmed by the presence of an appropriate ETCO_2_ curve in both the groups. The depth of the tracheal tube was confirmed using fiberoptic bronchoscopy. After induction of anesthesia, the radial artery was catheterized for continuous arterial pressure monitoring. An additional intravenous route for the fluid challenge was secured with a 16-gauge catheter or central venous catheterization as appropriate. After induction of anesthesia in both the groups, a thoracic epidural catheter was inserted at the T4–T6 level by thoracic anesthesiologists for postoperative pain control.

After patient positioning and confirmation of the tube position by fiberoptic bronchoscopy, pressure-controlled OLV was started with a peak inspiratory pressure (PIP) of 15 cmH_2_O and a positive end-expiratory pressure (PEEP) of 5 cm H_2_O. The respiratory rate varied from 8 to 17, maintaining an ETCO_2_ of 35–40 mmHg. When ETCO_2_ increased to >40 mmHg despite a respiratory rate of more than 17, PIP was increased by 1 cmH_2_O. When the partial pressure of arterial oxygen (PaO_2_) was <80 mmHg or SpO_2_ < 96%, PIP was also incrementally increased to 20 cmH_2_O. In case of requirement of PIP > 20 cmH_2_O, FiO_2_ was increased by 0.1. If patient’s oxygenation was clinically well maintained, FiO_2_ was decreased by 0.05–0.1 p (minimum 0.3). Intermittent recruit maneuver (RM) was not provided routinely during OLV except apparent hypoxia continues despite increase of FiO_2_ or PIP. After OLV, a brief RM using 30–40 cmH_2_O followed by PEEP was performed in both the groups. At the end of surgery, a high gas flow (more than 8 L/min) at 5 cmH_2_O of adjusted pressure limitation was used for patients’ tracheal extubation. In postanesthetic care unit (PACU), 2–4 L/min of oxygen was supplied to patients via nasal prong for 15 or 20 min.

### 2.5. Monitoring of Neuromuscular Blockade

Neuromuscular transmission was monitored by the response of the adductor pollicis muscle using acceleromyography (TOFscan^®^; Dräger, Lübeck, Germany) and the TOF count was measured at every 15 min after tracheal intubation. In both the groups, NMB was maintained with a continuous infusion of rocuronium bromide. In group M, rocuronium infusion was started after the appearance of TOF count of 1 and the infusion rate was titrated to maintain a TOF count of 1 or 2 (0.1–0.5). In group D, rocuronium infusion was started at 15 min after tracheal intubation and the infusion rate was titrated to maintain a PTC of 1 or 2 (0.5–1.0). When the NMB deepened in both the groups, continuous infusion of rocuronium was temporarily stopped and the measurement interval was changed to 5 min until the recovery of NMB. A rescue dose of 10 or 20 mg rocuronium bolus was administered intravenously when accidental diaphragmatic movement was observed or when the surgeon deemed it necessary. Subsequently, the rocuronium infusion was temporarily stopped and the measurement interval was changed to 5 min until the recovery of NMB. The rocuronium infusion was stopped after the end of OLV in both the groups. After the surgery was completed, patients in group M received 2 mg/kg of sugammadex (Bridion™, Merck, NJ, USA) and patients in group D received 4 mg/kg of sugammadex for NMB reversal. All patients were extubated after confirmation of TOF ratio >0.9 and transferred to the postanesthetic care unit (PACU) or the intensive care unit (ICU). To assess postoperative residual paralysis, TOF was measured immediately and at 15 min after admission to the PACU or the ICU.

### 2.6. Outcome Measurement

The ventilatory parameters including tidal volume, FiO_2_, PIP, P_plat_, dynamic compliance, and PEEP, hemodynamic parameters, and SpO_2_ were measured at every 15 min during OLV. Arterial blood gas analysis was routinely performed immediately after arterial catheterization, at 15 min after one/two lung ventilation, randomly as required during OLV, and immediately after transferring to the PACU or the ICU in all patients.

All patients underwent routine chest CT on postoperative day (POD) 2. CT acquisition and image analysis were performed as follows: chest CT was performed on a 256-slice revolution CT scanner (GE Healthcare, Milwaukee, WI, USA). Scanning parameters were 100 or 120 kVp, noise index of 20 with automatic tube current modulation, collimation of 0.625 mm, pitch of 1.105, gantry rotation time of 0.5 s, and matrix size of 512 × 512. All CT images were reconstructed with slice thickness of 1.25 mm and a standard kernel using adaptive statistical iterative reconstruction-V was used to calculate the atelectasis volume. CT images were retrospectively analyzed by a radiologist with more than five years of experience in thoracic radiology. The radiologist was blinded to the patients’ clinical information. We used commercially available software (Terarecon, San Mateo, CA, USA) to quantitatively measure the aerated and atelectatic lung volumes using threshold methods (Figure 1). Since the observer visually identifies the aerated or atelectatic lung, boundaries were automatically drawn after applying threshold methods by summation of predefined Hounsfield unit (HU) pixels. The volume was then calculated for the selected area. A predefined HU threshold was −100 to 1000 HU for the aerated lung and −100 to 100 HU for the atelectatic lung. The proportion of postoperative atelectasis was defined as atelectatic lung volume/total lung volume (atelectatic lung volume + aerated lung volume, mL).

The secondary outcomes were other PPCs including pneumonia, pleural effusion, pulmonary edema, and pneumothorax on chest CT. Pneumonia was defined as the presence of at least one definitive radiologic examination and at least one sign of pneumonia (fever, leukopenia, leukocytosis, or altered mentality with no other cause), as well as at least one microbiologic laboratory finding or at least two clinical symptoms, which was described in a previous study [9]. Pleural effusion, pulmonary edema, or pneumothorax was diagnosed directly on chest CT. To calculate the incidence of atelectasis as a component of PPC, the actual volume of atelectasis >4% in chest CT was used [10]. Additionally, intraoperative parameters during OLV such as dynamic lung compliance (C_dyn_), incidence of desaturation (SpO_2_ < 95%), FiO_2_ increase, the lowest value of arterial oxygen saturation (SaO_2_), and PaO_2_ were compared. Laboratory tests were performed before and after the surgery. The length of hospital stay and ICU stay were also recorded.

### 2.7. Statistical Analysis

Based on a previous study in which showed a 28.5% difference of overall PPCs by the intraoperative NMB depth in patients undergoing thoracic surgery [11], we calculated the minimal requirement (53 patients) per group to detect a 25% difference in the atelectasis volume, assuming a type I error of 0.5 and a desired power of 0.8 was used for the experimental design. Considering a possibility of 10% loss due to unexpected circumstances, we recruited 59 patients per group. Demographic and preoperative data were compared using the intent-to-treat analysis. Data regarding the incidence of postoperative complications were calculated using the per-protocol analysis, since the actual number of analyzed patients can affect the outcome values. Shapiro–Wilk test was used to test the normality of the data. Continuous data were analyzed using Student’s t-test or Mann–Whitney U test, depending on normality. Categorical data were analyzed using the chi-squared analysis or Fisher’s exact test when applicable. Repeated data were analyzed using repeated-measures analysis of variance. Statistical analyses were performed using a standard statistical program (MedCalc^®^; MedCalc Software, Ostend, Belgium). All values were expressed as mean ± standard deviation, median (interquartile range), or number (percentage). Statistical significance was set at *p* < 0.05.

## 3. Results

Altogether, 118 patients were enrolled and 59 patients were allocated to each group. A patient in group D declined to participate in the study after enrollment. Finally, 114 patients were analyzed after exclusion of one patient in group M and in two patients in group D due to missing postoperative CT data (Figure 2). Demographics and preoperative data are presented in Table 1. The patients’ demographics and preoperative laboratory test results did not show significant differences between the groups except the higher incidence of abnormal findings such as atelectasis, pleural effusion, and pneumothorax on preoperative chest radiography.

The intraoperative data are presented in Table 2. During OLV, group D showed higher C_dyn_ than group M (Figure 3, *p* = 0.006). In addition, mean C_dyn_ during OLV was higher in group D (26.6% in group D vs. 24.1% in group M, *p* = 0.026). During OLV, the incidence of FiO_2_ increase and SaO_2_ < 95% was similar between the groups.

The proportion of postoperative atelectasis did not show a significant difference between the groups (1.32 (0.47–3.20)% in group M and 1.41 (0.24–3.07)% in group D, *p* = 0.690, Figure 4). The actual atelectasis volume was 38.2 (12.8–61.4) mL in group M and 31.9 (7.8–75.0) mL in group D (*p* = 0.954). Other postoperative outcomes and laboratory test results are presented in Table 3. Overall PPCs and each PPC did not show significant differences between the groups. Although group D seemed to exhibit better immediate postoperative PaO_2_ and partial pressure of carbon dioxide, there were no statistically significant differences in all the laboratory tests between the groups. None of the patients experienced failure of thoracic epidural analgesia. None of the patients were re-intubated for respiratory failure after surgery.

## 4. Discussion

In the present study, both the proportion and the volume of postoperative atelectasis showed no differences between patients with deep and moderate NMB during thoracic surgery. Deep NMB provides higher C_dyn_ than moderate NMB and tends to reduce oxygen requirement during OLV. However, parameters regarding intraoperative hypoxemia, postoperative outcomes, and even overall incidence of PPCs did not show a significant difference between the groups. The present study was too small to find any impact of depth of NMB on outcome. Larger observational studies on other patient groups found less complications in patients getting deep NMB and it would be interesting to analyze this also in OLV patients [12].

The present study has been looking only to atelectasis as one the most frequent PPC and can therefore make no conclusion to other factors of PPC. There are several causes of postoperative atelectasis during thoracic surgery. During OLV, the ventilated lung can collapse due to gravity or surgical compression (compression atelectasis) [13]. The use of higher oxygen concentrations also contributes to intraoperative atelectasis (absorption atelectasis) [13,14]. Furthermore, the residual NMB effect can aggravate postoperative atelectasis [3]. The residual NMB effect may decrease the patient’s postoperative inspiratory effort, which may cause inappropriate expansion of the non-ventilated lung after surgery. Chest CT can be used to diagnose or to evaluate atelectasis postoperatively. It is valuable, as it can measure the whole and the regional lung volume quantitatively when compared with simple chest radiography [15]. In patients undergoing thoracic surgery who have preoperative pleural effusion or pneumothorax, quantitative assessment of postoperative atelectasis using chest CT enables an accurate and objective evaluation of atelectasis volume and provides clinical implications for the inspiratory effort of the patient after surgery or the residual NMB effect.

In our result, postoperative atelectasis did not show the significant difference between moderate and deep NMB. Although increased C_dyn_ may slightly prevent intraoperative lung volume reduction in the deep NMB group, its clinical significance seems negligible. Instead, achieving a greater than 0.9 TOF ratio for tracheal extubation using an appropriate dose of sugammadex would contribute to little difference between both groups. Recent studies have compared the incidence of PPCs and showed similar results between the deep and the moderate NMB groups [8,16]. However, maintaining intraoperative deep NMB may contribute to increased lung compliance [17]. Our results also may provide an evidence that deep NMB is helpful for the prevention of intraoperative reduction of lung volume during OLV by increased lung compliance [18]. Moreover, very few studies have quantitatively evaluated the amount of postoperative atelectasis using chest CT image reconstruction. In the present study, we quantitatively measured the whole lung volume and atelectasis volume to directly assess the effect of intraoperative depth of NMB on postoperative atelectasis.

Theoretically, despite providing surgical convenience or increasing C_dyn_ during surgery, deep NMB can prolong the recovery of NMB, thereby increasing the risk of residual paralysis [19]. The residual NMB effect inhibits upper airway muscle function more than respiratory muscle function inducing inspiratory obstructive breathing with strong negative intra alveolar pressures inducing atelectasis and lung edema [20]. However, sugammadex, a new reversal agent that can selectively bind with rocuronium, provided faster recovery and reduced the risk of residual paralysis dramatically in deep NMB when compared with neostigmine [21]. In the present study, 4 mg/kg sugammadex in the deep NMB group could provide rapid and effective reversal of neuromuscular function. Consequently, none of the patients showed postoperative residual neuromuscular paralysis in the PACU or in the ICU. Although maintaining deep NMB during surgery may increase the risk of residual paralysis, appropriate intraoperative neuromuscular monitoring and use of sugammadex may improve spontaneous respiratory movement to achieve complete re-expansion of the lung after surgery. As shown in our results, no difference in postoperative atelectasis between deep and moderate NMB group supported a clinical evidence that deep NMB can be used safely perioperatively.

Interestingly, despite the similar results between the groups, the absolute values of postoperative atelectasis were relatively small in both the groups. We believe that appropriate lung protective ventilation, optimal NMB status monitoring strategy and providing thoracic epidural analgesia contributed to this finding. Although we did not strictly follow the standard lung protective ventilatory guideline in the present study [4], the ventilatory strategy including low tidal volume, PEEP, and low FiO_2_ during extubation would contribute to decrease the postoperative atelectasis. Moreover, we monitored the NMB status at every 15 min during the surgery and at every 5 min when rescue rocuronium was administered or the infusion rate was changed. This strategy was helpful in maintaining a stable NMB status intraoperatively and in ensuring appropriate dose of sugammadex at the end of surgery. In addition, we administered thoracic epidural analgesia in both groups after induction of anesthesia what allowed to avoid giving opioids and sedatives postoperative reducing obstructive breathing as well. Thoracic epidural analgesia can effectively decrease postoperative pain and reduce further postoperative atelectasis by enabling deep breathing, coughing, and exercise [22]. Thoracic epidural analgesia was safely administered by a staff anesthesiologist specializing in thoracic anesthesia. In the present study, no patients experienced a failure of thoracic epidural analgesia postoperatively.

The present study has some limitations. First, there was a significant difference in the sex and the incidence of abnormal findings on preoperative chest radiography between group M and group D. Thus, the randomization was not well achieved. However, it is unknown how sex could affect postoperative atelectasis. Furthermore, abnormal findings on preoperative chest radiography were correlated with the patients’ diagnoses and were clinically well controlled. We compared the volume and the proportion of postoperative atelectasis directly and not the overall incidence of PPCs itself. All patients were individually evaluated before surgery, and there were no patients with compromised cardiopulmonary function. Second, although it is a new NMB monitoring device using acceleromyography and shows a good agreement with the conventional device, the TOFscan^®^ which was used in the present study can have a limitation of use compared to TOF-watch because it does not perform a baseline calibration and signal stabilization [23]. Third, we did not evaluate PPCs after POD 2. The primary outcome (the proportion and the volume of postoperative atelectasis) was measured using CT on POD 2. Although radiography evaluation on POD 2 is standard protocol in our institution, PPCs such as postoperative pneumonia and pulmonary edema can occur until 1–2 weeks after surgery. We used CT on POD 2 to evaluate atelectasis and other PPCs such as pneumonia or pulmonary edema but did not follow up patients after that. In the present study, we focused more on the quantitative measurement of postoperative atelectasis than the other general outcomes.

In conclusion, a quantitatively measured postoperative atelectasis did not show a difference between patients maintaining deep and moderate NMB during thoracic surgery with OLV when full reversal was verified before extubation.

## Figures and Tables

**Figure 1 jcm-10-03228-f001:**
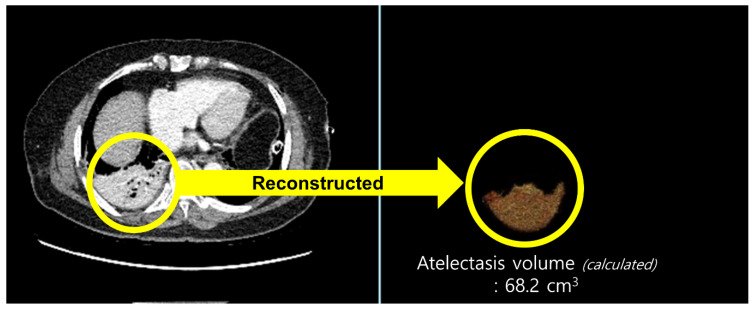
Quantitatively measured volume of an atelectatic lung using threshold methods in chest computed tomography. For the calculation of atelectatic lung volume, the area with predefined Hounsfield unit (HU) threshold of −100 to 100 HU was selected in the chest computed tomography images (yellow circle on the left) and reconstructed (yellow circle on the right).

**Figure 2 jcm-10-03228-f002:**
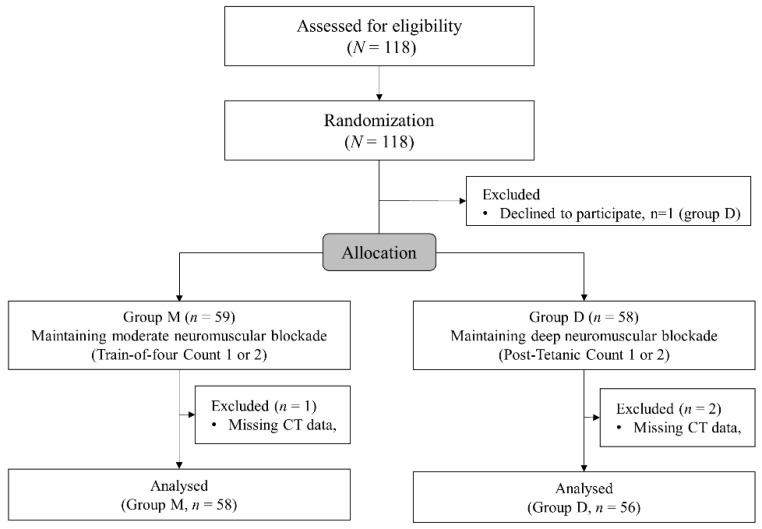
The CONSORT flowchart.

**Figure 3 jcm-10-03228-f003:**
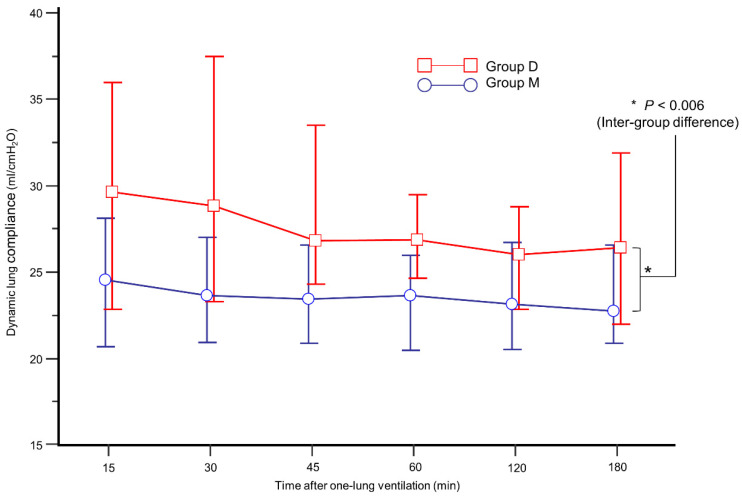
Comparison of mean dynamic lung compliance during one-lung ventilation between group M (blue circle) and group D (red box).

**Figure 4 jcm-10-03228-f004:**
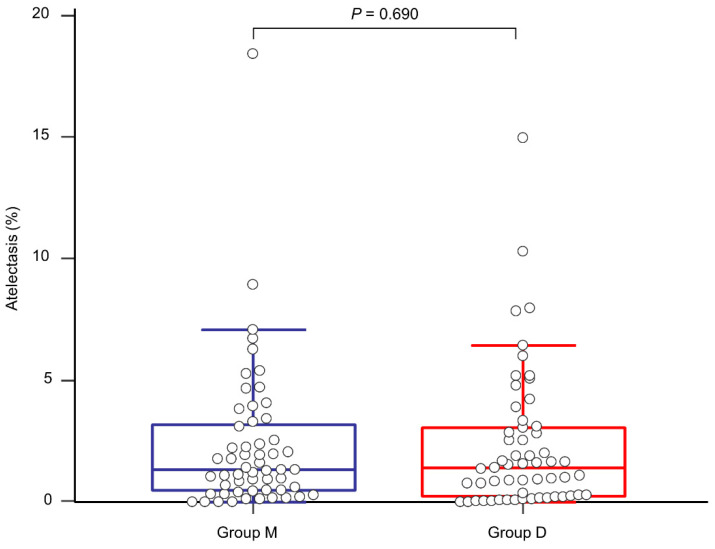
Comparison of the proportion of postoperative atelectasis between group M (blue line) and group D (red line).

**Table 1 jcm-10-03228-t001:** Demographics and surgical data.

Variables	Group M (*n* = 59)	Group D (*n* = 58)	*p*
Age	64 (59–71)	64 (58–73)	0.611
Sex, male/female	29 (49.2)/30 (50.8)	43 (74.1)/15 (25.4)	0.005
ASA physical status 1	8 (13.6)	7 (11.9)	0.910
2	44 (74.6)	46 (78.0)	
3	7 (11.9)	6 (10.2)	
Height (cm)	161 (152.0–165.7)	165 9 (158.7–169.4)	0.006
Weight (kg)	62 (56.5–71.6)	62.8 (53.9–69.3)	0.844
Body mass index (kg/m^2^)	24.3 (22.2–26.3)	23.6 (20.2–25.8)	0.064
Preoperative abnormal chest radiography	2 (3.4%)	16 (27.1)	0.023
Pleural effusion	0 (0.0)	8 (13.6)	
Pneumothorax	2 (3.4)	4 (6.8)	
Others (Pulmonary edema or infiltration, atelectasis, hemothorax)	0 (0.0)	4 (6.8)	
Type of surgery			0.070
Wedge resection	6 (10.2)	9 (15.3)	
Segmentectomy	12 (20.3)	7 (11.9)	
Lobectomy	33 (55.9)	29 (49.2)	
Lobectomy + Wedge resection	2 (3.4)	0 (0.0)	
Lobectomy + Segmentectomy	0 (0.0)	1 (1.7)	
Pneumonectomy	0 (0.0)	1 (1.7)	
Decortication	1 (1.7)	9 (15.3)	
Etc. (including Pleura, Mediastinum, Esophagus)	5 (8.5)	3 (5.1)	
Incidence of open conversion	7 (11.9)	11(19.0)	0.308

All data are expressed as median (interquartile range) or number (%). ASA: American Society of Anesthesiologists.

**Table 2 jcm-10-03228-t002:** Intraoperative data.

Variables	Group M (*n* = 59)	Group D (*n* = 58)	*p*
Total anesthesia time (min)	260 (220–316)	250 (205–331)	0.799
Total operation time (min)	200 (136–243)	183 (132–241)	0.524
One-lung ventilation time (min)	172 (125–220)	165 (125–217)	0.787
Parameters during one-lung ventilation *			
Mean tidal volume (mL)	292 (196–386)	302 (214–428)	0.122
Mean respiratory rate (/min)	15 (14–16)	15 (15–16)	0.723
Mean positive end-expiratory pressure (cmH_2_O)	5 (5–5)	5 (5–5)	0.950
Mean lung compliance (mL/cmH_2_O)	24.1 (21.8–27.2)	26.6 (23.5–30.6)	0.026
PaO_2_, lowest (mmHg)	85.7 (72.7–105.8)	81.9 (71.8–97.4)	0.347
SaO_2_, lowest (mmHg)	94.6 (93.0–96.5)	94.1 (92.8–96.1)	0.663
Incidence of FiO_2_ increase (>0.5)	21 (35.6)	12 (20.3)	0.167
Incidence of SaO_2_ < 95%	34 (58.6)	36 (61.0)	0.792
Incidence of additional NMBA administration	23 (39.0)	22 (37.3)	0.850
Incidence of conversion to open surgery	7 (11.9)	11 (18.6)	0.308
Total amount of administered propofol (mg)	1359 (1090–1756)	1413 [1056–1681)	0.771
Total amount of administered remifentanil (µg)	1591 (1184–2132)	1508 [1100–2030)	0.333
Total amount of administered crystalloid (mL)	1200 (880–1495)	1300 [900–1760)	0.314
Total amount of transfused RBCs (unit)	0 (0–0)	0 [0–0)	0.172
Urine output (mL)	390 (250–508)	320 [230–500)	0.302

All data are expressed as median (interquartile range) or number (%). PaO_2_: partial pressure of arterial oxygen, SaO_2_: arterial oxygen saturation, FiO_2_: inspiratory fraction of oxygen, NMBA: neuromuscular blocking agent, RBCs: red blood cells. * Parameters during one lung ventilation including the mean tidal volume, respiratory rate, positive end-expiratory pressure, and lung compliance indicate the median and interquartile range of the mean values of each patient’s parameter.

**Table 3 jcm-10-03228-t003:** Postoperative data.

Variables	Group M (*n* = 58)	Group D (*n* = 56)	*p*
Postoperative pulmonary complications			
Overall	43 (74.1)	42 (73.7)	0.831
Pleural effusion			0.779
Ipsilateral	19 (32.8)	21 (36.8)	
Contralateral	2 (3.4)	4 (7.0)	
Bilateral	15 (25.8)	12 (21.1)	
Pneumothorax	4 (6.9)	5 (8.8)	0.730
Pulmonary edema	0 (0)	1 (1.8)	0.317
Pneumonia	1 (1.7)	2 (3.5)	0.560
Postoperative arterial blood gas analysis			
pH	7.352 (7.305–7.377)	7.361 (7.326–7.399)	0.135
PaO_2_ (mmHg)	133.0 (97.8–172.0)	156.5 (103.4–198.5)	0.085
PaCO_2_ (mmHg)	44.0 [40.7–47.7)	42.1 (38.7–46.5)	0.087
HCO_3_^−^(mEq/L)	23.5 (22.4–25.1)	23.3 (21.4–25.2)	0.520
SaO_2_ (%)	97.7 (97.5–98.4)	98.1 (97.1–98.5)	0.149
Intensive care unit stay (days)	1 (1–1)	1 (1–1)	0.905
Hospital stay (days)	11 (9–16)	12 (9–16)	0.668

All data are expressed as median [interquartile range] or number (%). PaCO_2_: partial pressure of arterial carbon dioxide, PaO_2_: partial pressure of arterial oxygen, SaO_2_: arterial oxygen saturation.

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
