# Peer review of "Effect of Deep versus Moderate Neuromuscular Blockade on Quantitatively Assessed Postoperative Atelectasis Using Computed Tomography in Thoracic Surgery; a Randomized Double-Blind Controlled Trial"

_jcm, 2021, doi:10.3390/jcm10153228_

Round 1

Reviewer 1 Report

Comment to the authors

Effect of deep NMB versus moderate NMB on atelectasis

Interesting study but requires conduct according to LPV guidelines to be sure that atelectasis is due to an effect of depth of NMB and no due to misconduct of ventilation management.

  1. Therefore please explain what steps of the LPV are followed what steps have been overlooked.See therefore article from Young 2019  and a short overview below:

Young, et al. BJA 2019; 123 (6): 898-913

The essential LPV guidelines for every adult patient.

Induction of anesthesia

  1. Always beach chair (20°) during induction; (avoid flat supine position during induction in every patient if not contra indicated).
  2. Use always CPAP prior to the loss of spontaneous ventilation. (use PSV with zero support or use high flow an APL at 5 if not contra indicated)
  3. Monitor during induction for an obstructive breathing pattern and use a combination of appropriate techniques. (Guedel, head position, more CPAP with good fitting mask or nasal mask only)

Maintenance of anesthesia

  1. After induction (with FiO2 80%) start with FiO2 4. Thereafter, use the lowest possible FiO2 to achieve SpO2 ≥ 94%.
  2. The ventilator should be set to deliver VT 6-8 mL/kg LBW with PEEP minimal 5 cmH2O. (never zero peep is possible, also in spont breathing or LMA) Higher PEEP may be required in obese patients, during pneumoperitoneum, and during prone or Trendelenburg
  3. The lowest aggressive ventilation mode is probably the best. As pressure support ventilation has never been proved to be better we should stay with volume controlled and probably insufflate as slowly as possible using I/E ratio of 1/1 and the lowest inspiratory flow possible. (more data is required to recommend this setting as being superior)
  4. Dynamic compliance or driving pressure (PPlat-PEEP) should be monitored in every patient. Decreasing compliance should be treated with recruitment (RMs) combined with sufficient PEEP. RMs should be performed using the lowest effective PPlat (30-40 cmH2O in non-obese, 40-50 cmH2O in obese) and shortest effective time or fewest number of breaths. (The bag-squeezing RM should be avoided in favor of a ventilator-driven RM).
  5. After lung collapse due to suctioning, one lung ventilation, tube disconnection, PEEP interruption or pneumoperitoneum perform always a RM followed by PEEP and verify normalization of dyn lung compliance.

Emergence from anesthesia

  1. Always extubate under CPAP. (use PSV with zero support or use high flow and APL at 5) Avoid ETT suctioning immediately prior to tracheal extubation. (apply RM before extubation if suctioning was needed).
  2. Low FiO2 (<0.4) during emergence from general anesthesia can improve pulmonary function in the postoperative period. (When high FiO2 (> 0.8) is used during emergence, use CPAP with Fi02 (<0.3) to reduce the risk of resorption atelectasis in PACU)
  1. Avoid PORC (verify full reversal to TOF > 90%), avoid opioids (use opioid free anesthesia or low opioid) and sedatives. If opioids are used intra operative or postoperative apply CPAP with low FiO2 as also requested in OSAS patients.

Postoperative care

  1. Avoid routine application of supplemental oxygen without investigating and treating the underlying cause. Postoperative oxygen is recommended when room air Sp02 falls below 94%.

If several steps of the LPV are not followed in this study repeat this study or explain that due to the conduct without LPV (as most anesthesiologist are still performing)  no difference could be found.  But what is then the value of this study as no difference does not help us either…

  1. No other study did investigate yet the relationship between deep NMB and atelectasis as we do not suspect that there would be any relationship in general or in any special subgroup.  Why is therefore this study setup in OLV patients before evaluating this in general if you suspect that this general believed aspect is not true.

Therefore give explanation why deep NMB might have an impact on atelectasis.  Do you have data,  clinical experience  or older publications hinting to this or do you just want to strength that it is useless to give deep NMB for this purpose. ( are some anesthesiologists claiming this?)

Deep NMB will decrease lung compliance a little as you found also but this was not the purpose of this investigation.  Deep NMB might change the thorax rigidity and therefore you will see this difference and if measuring transpulmonary pressures no difference should be found… but this last is not your research question..

Author Response

Thank you for the kind and detailed comments. According to your comments, we revised the manuscript and marked it in red color. Please, see the attachment. 

Reviewer 2 Report

Journal of Clinical Medicine

Effect of deep versus moderate neuromuscular blockade on quantitatively assessed postoperative atelectasis using computed tomography in thoracic surgery, a randomized double-blind controlled trial.

From Bong-Jae Lee, et al.

This paper demonstrates that intraoperative deep neuromuscular blockade did not affect postoperative atelectasis when compared with moderate neuromuscular blockade.

The Authors cite as previous studies the ref 8 and 10, that are from other groups. “Accordingly, to” sounds better and a brief summary of the definition (ref 8) or statistics (ref 10) could help the reader.

Table should be improved thank to the use of abbreviations in the 1st column (and appropriate legend at the bottom of the table). In the table 1 the Authors use a legend at the top of the table, that should do at the bottom. They use AST and ALT and BUN, which are missing in their tables. In table 2 the authors report the median of the mean tidal, mean RR and of the mean positive end expiratory pressure and lung compliance. These are supposed to be the median and interquartile range of the mean values of the single patient. The Authors should clarify in the legend. Abbreviations are strongly suggested to contain in a column the parameters name. HCO3 is in mEq/L?

Tables are different in word style from text.

Lines 242-245 some useless parameters are repeated.

Some refs are heterogenous in style, please the Authors to check n. 7.

Author Response

Thank you for the kind comments. According to your recommendations, we revised the manuscript and marked it in red color. Also, we responded to your comments line by line. Please, see the attachment. 

Round 2

Reviewer 1 Report

has improved a lot and need minor adaptations as shown in added word doc in red. 

congratulation, message is needed for clinical practice when focus on 

"no difference in atelectasis incidence if full reversal verified before extubation"

Author Response

Thanks for the kind and detailed comment of the Reviewer 1. Please, see the attachment. 
